# Nucleophilic Radiofluorination Using Tri-*tert*-Butanol Ammonium as a Bifunctional Organocatalyst: Mechanism and Energetics

**DOI:** 10.3390/molecules27031044

**Published:** 2022-02-03

**Authors:** Young-Ho Oh, Sandip S. Shinde, Sungyul Lee

**Affiliations:** 1Department of Applied Chemistry, Kyung Hee University, Deogyeong-daero 1732, Yongin-si 17104, Gyeonggi-do, Korea; chem_yhoh@daum.net; 2Department of Nuclear Medicine, Molecular Imaging and Radiochemistry, Friedrich-Alexander University Erlangen-Nürnberg (FAU), 91054 Erlangen, Germany

**Keywords:** ^18^F-fluorination, tri-*tert*-butanol ammonium, organocatalysis, mechanism

## Abstract

We present a quantum chemical analysis of the ^18^F-fluorination of 1,3-ditosylpropane, promoted by a quaternary ammonium salt (tri-(*tert*-butanol)-methylammonium iodide (TBMA-I) with moderate to good radiochemical yields (RCYs), experimentally observed by Shinde et al. We obtained the mechanism of the S_N_2 process, focusing on the role of the –OH functional groups facilitating the reactions. We found that the counter-cation TBMA^+^ acts as a bifunctional promoter: the –OH groups function as a bidentate ‘anchor’ bridging the nucleophile [^18^F]F^−^ and the –OTs leaving group or the third –OH. These electrostatic interactions cooperate for the formation of the transition states of a very compact configuration for facile S_N_2 ^18^F-fluorination.

## 1. Introduction

The ^18^F-fluorination [1,2,3,4,5,6,7,8,9,10,11,12,13,14,15,16,17,18,19,20] of organic compounds is gaining considerable importance for synthesizing chemicals that can be employed as radiotracers for the diagnosis of various diseases [21,22,23] by the highly sensitive imaging technique of positron emission tomography (PET) [24,25]. Electrophilic substitution reactions [26,27,28] using the carrier added [^18^F]F_2_ gas were the earlier method for this purpose, but this approach usually suffers from poor radiochemical yields (RCYs) and from the difficulty of handling the [^18^F]F_2_ gas. The nucleophilic incorporation of [^18^F]fluoride may be considered as a favorable alternative, but so far, this approach mainly requires azeotropic drying and multistep synthetic routes. Generally, this is undesirable for radiofluorination, as ^18^F possesses a relatively short half-life of 109.77 min. Various non-radioactive nucleophilic fluorination methods [29,30,31,32,33,34,35] have been developed using various phase-transfer catalysts (PTCs) for their application in radiopharmaceutical chemistry in which hydrogen bonding between protic PTC and nucleophile [^18^F]fluoride enhances the reaction and selectivity [36,37]. The recent development of tri-(*tert*-butanol)-methylammonium [^18^F]fluoride (TBMA-^18^F) [13] as a promoter for radiofluorination demonstrated an excellent [^18^F]fluoride elution efficiency under the reduced basicity of [^18^F]fluoride, with moderate to good obtained RCYs. This technique seems to allow a single-step reaction scheme for the synthesis of ^18^F-labeled substances, which would be a significant advance. Shinde and co-workers proposed that these excellent organocatalytic properties of the counter-cation may be attributed to the coordinating capacity of the two functional groups (ammonium and hydroxyl) in TBMA^+^. This rate enhancement, promoted by the counter-cation, which would be the first such example for ^18^F-fluorination, exhibits significant advantages over more conventional methods using alkali metal cations, M^+^ (M = K, Rb, Cs), because in the latter case the Lewis base promoters such as bulky alcohols [8,32], oligoethylene glycols [8,31,32], ionic liquids [8,33], or crown ethers [34,35] are required to suppress the harmful Coulomb forces of the counter-cation on the nucleophile.

Here we elucidate the mechanism of the rate enhancement of ^18^F-fluorination by a quaternary ammonium salt (tri-(*tert*-butanol)-methylammonium iodide (TBMA-I)) (Figure 1) by quantum chemical methods. We present the analysis of the ^18^Ffluorination of 1,3-ditosylpropane (**1**), finding that the counter-cation TBMA^+^ acts as a bifunctional promoter: the –OH groups in TBMA^+^ act as a bidentate ‘anchor’ by forming hydrogen bonds with the nucleophilic [^18^F]fluoride and the tosylate (–OTs) leaving group or the third –OH. These electrostatic interactions cooperate for the formation of the transition state (TS) of a very compact configuration for facile S_N_2 ^18^F-fluorination. The fluorinating agent TBMA-^18^F reacts with substrate **1** as a contact ion-pair, in which the counter-cation TBMA^+^ and the nucleophile [^18^F]F^−^ are in close contact.

## 2. Results

Figure 1 presents the structure of TBMA-I, which is transformed to TBMA-^18^F by Shinde and co-workers’ procedure [13], along with the ^18^F-fluorination of the model substrate 1,3-ditosylpropane **1**. Carrying out the reaction in CH_3_CN selectively produced the ^18^F-labeled product **[^18^F]2** with an RCY of 20% in CH_3_CN (Table 1). In Shinde and co-workers’ experiments, TBMA-^18^F is recovered from QMA cartridges after treatment with methanolic TBMA-I, and the counter-cation TBMA^+^ accelerates the ^18^F-fluorination, thus the reaction given in Figure 1 would be the key process.

We carried out density functional theory (DFT) calculations for the reaction in CH_3_CN and obtained the reaction mechanism and energetics shown in Figure 1 and Figure 2, in which the TSs (structures of pre- and post-reaction complexes and Cartesian coordinates are listed in Appendix A, Appendix A) and the energy profile are depicted.

We found three S_N_2 mechanisms with different TSs and energetics (Figure 1). In all TSs (TS-S_N_2_1, TS-S_N_2_2, TS-S_N_2_3), the ammonium TBMA^+^ and the nucleophile [^18^F]F^−^ are in close contact (*R*_N-F_ = 3.568–3.735 Å), indicating that the reaction proceeds via the contact ion-pair (CIP) mechanism. In these TSs, hydrogen bonding between the two hydroxyl –OHs, the third –OH, or the tosylate (–OTs) leaving group are shown to play a crucial role in the reaction mechanism. Two of the –OHs bind to the nucleophile [^18^F]F^−^ in all TSs. The role of the 3rd –OH, however, is somewhat different: in TS-S_N_2_1, it interacts with the leaving group –OTs, assisting their departure, whereas in two other TSs, it forms a hydrogen bond with –OH. These electrostatic interactions help to form compact pre-reaction complexes and TSs (see Appendix A, Appendix A) and TSs that are favorable for S_N_2 fluorination. Thus, the two –OH groups act as a bidentate ‘anchor’, and the counter-cation TBMA^+^ plays its role as a promoter for the ^18^F-fluorination by providing these two –OHs. The structures of the three TSs in Figure 1 are similar, the difference being in the relative orientation of the –OTs leaving group and TBMA^+^.

The relative feasibility of the three CIP mechanisms may be assessed by invoking the Curtin–Henderson principle [38]: if the transformation from free reactants to the pre-reaction complex is irreversible (that is, no equilibrium between the two), the reaction path with the lowest Gibbs free energy of the TS (that is, with lowest Gibbs free energy of activation *G*^‡^ with respect to the free reactants) is favored, irrespective of the Gibbs free energies of the pre-reaction complexes along the reaction routes. The calculated *G*^‡^ for the three reaction routes are 31.3, 31.6, and 31.6 kcal/mol (Figure 2), suggesting that they may contribute almost equally (the difference of 0.3 kcal in *G*^‡^ for the case 1 and case 3 routes amounts to a ~1:1.5 ratio of reaction rates at the reaction temperature of 85 °C).

It would be interesting to compare the mechanism of ^18^F-fluorination by TBMA-^18^F shown here with the mechanism [39] of promotion of (non-radiative) fluorination by [bmim][F] in a solvent-free environment, experimentally demonstrated by Magnier and co-workers [40]. In the latter process, the acidic H-atom in the counter-cation bmim^+^ facilitates the detachment of the leaving group, whereas in the present case it is the –OHs interacting with the leaving group. It is to be noted that in both reactions the fluorinating agents (TBMA-^18^F, [bmim][F]) act as promoters for the reactions as well. Strong Coulombic attraction between the nucleophile and the counter-cation forces them to react as a CIP.

The relative Gibbs free energies of the pre-reaction complexes in Figure 2, with respect to those of free reactants, are worth mentioning: the (stationary) pre-reaction complexes are predicted to lie higher than free reactants, in contrast with most S_N_2 reactions we have studied. Although some theoretical studies [41,42,43] have suggested that pre-reaction complexes could not be obtained in some cases, the origin of the ‘well-skipping’ S_N_2 reactions predicted here is not clear.

One of the salient features of Shinde and co-workers’ procedure is that it produced the S_N_2 ^18^F-labeled product [^18^F]2 exclusively, without E2 elimination [44] products, and this observation is worth mechanistic scrutiny. This aspect of Shinde’s method is very important, because the exclusion of E2 products may be much more significant than the S_N_2 yield itself, especially for radiofluorination. Additional purification steps would be critically harmful when using the radiofluorinated product for clinical/diagnostic purposes due to the short lifetime of ^18^F. Figure 3 and Figure 4 present the two TSs (structures of pre- and post-reaction complexes and Cartesian coordinates are listed in Appendix A, Appendix A) and the corresponding energetics, respectively, of the E2 elimination of **1** in CH_3_CN. The Gibbs free energies of activation *G*^‡^ for the elimination process are calculated to be 34.1~35.5 kcal/mol (Figure 4), which are considerably higher (by 2.8~4.2 kcal/mol) than those for the S_N_2 ^18^F-fluorination (Figure 2), accounting for the experimentally observed exclusive formation of S_N_2 products. Thus, it seems that although the hydrogen bonding between the two –OHs and the nucleophile [^18^F]F^−^ may retard the S_N_2 fluorination to a certain degree, these electrostatic interactions may also promote the reaction by facilitating the formation of a pre-reaction complex and TS that are much more advantageous for the S_N_2 over the E2 process.

## 3. Computational Details

The M06-2X/6-311G** method [45,46] was employed as implemented in Gaussian16 [47], which proved to be very instructive for treating non-covalent interactions [48]. We adopted the cluster/continuum approximation, including the effects of the solvent continuum by the SMD-PCM method [49]. The reference for the zero of the free energy is taken as the ‘free’ reactant, for which the substrate and TBMA-^18^F are separated from each other in the solution phase. We carried out an extensive (but not exhaustive) search for stationary states over the potential energy surface of the system. Pre-reaction and post-reaction complexes were obtained by verifying that all harmonic frequencies are real. Transition states were obtained by ascertaining the imaginary frequency of the reaction coordinate, and also by performing the intrinsic reaction coordinate analysis.

## 4. Conclusions

We have presented the quantum chemical analysis to account for the activation of [^18^F]fluoride by tri-(*tert*-butanol)-methylammonium. The role of the –OHs in TBMA^+^ as a bifunctional organocatalyst has been manifested: the hydroxyl groups function as a bidentate ‘anchor’ to position the nucleophile [^18^F]F^−^ in a configuration suitable for the nucleophilic attack through hydrogen bonding. The elucidation of the underlying mechanism of the present S_N_2 processes would help to drive the development of this technique forward, using better ammonium counter-cations (for example, those with other alcohol moieties containing –OHs, such as –CH_2_OH, instead of –*t*-BuOH) and schemes (for example, using a less bulky leaving group such as a mesylate, to have more beneficial interactions between the counter-cation and leaving group) for more efficient ^18^F-fluorination. The application of the present methodology to the nucleophilic radiofluorination of a variety of radiotracers would be highly desirable.

## Data Availability

Not applicable.

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
