# Peer review of "Nucleophilic Radiofluorination Using Tri-tert-Butanol Ammonium as a Bifunctional Organocatalyst: Mechanism and Energetics"

_molecules, 2022, doi:10.3390/molecules27031044_

Round 1

Reviewer 1 Report

The present study is a theoretical/computational analysis of the fluorination reaction involving tri-tert-butanol ammonium as a counter cation of the fluoride ion. The reaction probably takes place via ion pair. The authors argue that the hydroxyls play a key role in promoting the reaction. However, the manuscript is too short and the analysis needs be improved before the work to be accepted.

  • The geometry of the free reactant was not reported. For this highly flexible molecule, a previous conformational search needs be done carefully.
  • I have verified that there are conformations with two hydrogen bonds to the fluoride, and the third hydroxyl can make an internal hydrogen bond to another hydroxyl. This structure seems be much stable. The author should include such structure in the conformational analysis of the free reactant.
  • In the transition states, only one hydroxyl was doing hydrogen bond. The another two are also active and a better search for more stable transition states needs be done. The authors written “Hydrogen bonding between the hydroxyl –OH and the tosyl-83 ate (-OTs) leaving group, and that between –OH and the nucleophile 18F- are shown to play critical role in the reaction mechanism. One of the OHs interacts with –OTs assisting its departure, whereas another forms a hydrogen bond with 18F-.” I do not see any hydrogen bonding with the leaving tosyl group in Figure 1.
  • More insights on the reactivity are needed. For example, what about the reactivity of tetramethylammonium fluoride compared with this new structure? Such comparison could provide more insights on the role of hydrogen bonding.

Suggested structure:

N         -6.55838        0.00180       -0.34405
C         -5.17628       -0.61270        0.12577
C         -4.23381       -1.32936       -0.91548
H         -5.40448       -1.29532        0.95474
H         -4.64075        0.24337        0.55768
O         -4.22162       -0.69866       -2.19043
C         -4.47776       -2.83916       -1.05573
H         -3.67836       -3.30127       -1.64825
H         -5.39603       -3.08916       -1.58694
H         -4.50087       -3.33221       -0.07823
C         -2.79255       -1.16936       -0.38401
H         -2.51395       -0.11066       -0.32097
H         -2.06807       -1.63133       -1.06533
H         -2.67405       -1.62042        0.60697
H         -4.92020       -1.08911       -2.79180
C         -6.22131        1.06644       -1.44988
C         -7.02752        2.40222       -1.55076
C         -7.42386       -1.18987       -0.85112
C         -8.79310       -0.96188       -1.55486
C         -9.90522       -0.46369       -0.62650
C         -9.27793       -2.31788       -2.11427
O         -8.65896       -0.06858       -2.66621
C         -6.59239        3.44786       -0.51640
O         -8.43281        2.22178       -1.41683
C         -6.78413        3.00182       -2.94865
H         -6.29054        0.54966       -2.39657
H         -5.17212        1.34902       -1.30403
H         -6.92465        3.20247        0.49396
H         -7.05111        4.41985       -0.73633
H         -5.50602        3.58276       -0.50849
H         -5.72288        3.21217       -3.11794
H         -7.34988        3.93139       -3.08140
H         -7.13175        2.32023       -3.73361
H         -8.69257        1.55675       -2.10017
H         -7.56117       -1.85315        0.01300
H         -6.78909       -1.71618       -1.54473
H         -7.99550       -0.50191       -3.28482
H         -9.98468       -1.08493        0.27181
H        -10.87389       -0.48909       -1.14107
H         -9.78426        0.57273       -0.31909
H         -9.46296       -3.03964       -1.31128
H         -8.54977       -2.75879       -2.80155
H        -10.20186       -2.19385       -2.69139
C         -7.17761        0.60509        0.89727
H         -6.44906        1.27349        1.36408
H         -7.41316       -0.20317        1.59744
H         -8.08506        1.14761        0.65704
F         -6.40603       -1.34992       -3.87884

Author Response

We thank the Reviewers for many helpful comments.

Reviewer 1

  1. The geometry of the free reactant was not reported. For this highly flexible molecule, a previous conformational search needs be done carefully.

We really thank the Reviewer for suggesting this. Indeed, we carried out calculations for the free reactant (TBMA+F-) described by the Reviewer, and found that its Gibbs free energy is lower than that given in the previous version of the manuscript. The structure is depicted in Figure 1, along with the updated TSs.

  1. The calculated structure of I have verified that there are conformations with two hydrogen bonds to the fluoride, and the third hydroxyl can make an internal hydrogen bond to another hydroxyl. This structure seems be much stable. The author should include such structure in the conformational analysis of the free reactant.

We updated the TSs with such interactions (Figure 1).

  1. In the transition states, only one hydroxyl was doing hydrogen bond. The another two are also active and a better search for more stable transition states needs be done. The authors written “Hydrogen bonding between the hydroxyl –OH and the tosyl-83 ate (-OTs) leaving group, and that between –OH and the nucleophile 18F- are shown to play critical role in the reaction mechanism. One of the OHs interacts with –OTs assisting its departure, whereas another forms a hydrogen bond with 18F-.” I do not see any hydrogen bonding with the leaving tosyl group in Figure 1.

In our previous version, only one –OH was depicted with the O atom in explicit form. We now revised the Figures so that the two hydrogen bonding between the two –OHs and F- are to be more easily noticed.

  1. More insights on the reactivity are needed. For example, what about the reactivity of tetramethylammonium fluoride compared with this new structure? Such comparison could provide more insights on the role of hydrogen bonding.

Tetraalkylammonium fluorides (for example, tetrabutylammoniumF) have usually been used for 19F fluorination before the methodology using metal salts was developed. These salts were okay, but were observed to suffer from significant amount of E2 products. For radiofluorination reactions, using quarternary ammonium fluorides may cause additional problems, as pointed out in Shinde’s paper. Indeed, the exclusion of E2 products is immensely important for radiofluorination, because additional purification step is critically harmful for using the reaction product for clinical/diagnostic purposes. Therefore, comparison with the fluorination by tetramethylammonium fluoride does not seem to be much instructive, unless for 19fluorination. On the other hand, there certainly exists a room for improvement in promoting efficiency since the reaction yield was only 20 % using TBMA+F-. The reaction rate is not large enough for appreciable reaction yield before the radioactive decay of 18F-. The role of –OHs in the substrate/promoter in Shinde’s experiments rather seems to be producing the pre-reaction complexes and TSs that are much more favorable for SN2 than E2 reactions. Thus, we add the following two paragraphs (pp. 5):

This aspect of Shinde’s method is very important, because the exclusion of E2 products may be much more significant than the SN2 yield itself especially for radiofluorination. Additional purification step would be critically harmful for using the radiofluorinated product for clinical/diagnostic purposes due to short lifetime of 18F.

Thus, it seems that although the hydrogen bonding between the two –OHs and the nucleophile [18F]F- may retard the SN2 fluorination to a certain degree, these electrostatic interactions may also promote the reaction by facilitating the formation of pre-reaction complex and TS that are much more advantageous for the SN2 over the E2 process.

Reviewer 2 Report

This work addresses an interesting question concerning the synthesis of 18F-fluorinated radiotracers. In particular, the authors look to elucidate the mechanism of rate enhancement of [18F]Fluorination by quantum chemical methods.

With that objective, the authors performed DFT calculations using the M06-2X functional (with 6-311G** basis set), carrying out extensive search for stationary states over the potential energy surface of the system, and compare the energy profiles for three possible SN2 mechanisms, and for three possible E1 mechanisms.

The work has merits, and may give a significant contribution to the field. There are, however, some questions that must be properly answered before the manuscript is considered for publication:

1- The choice of M06-2X functional should be justified. This functional may be the best for this study of reaction mechanisms, but the authors must refer to at least one of the several benchmarking studies to justify their choice and set the limits of accuracy (see also comment 6 below).

2 - For the quantum chemical description of anions, the presence of diffuse functions is often imperative. Is not the case for F-? The authors should state why.

3 - The authors state “We carried out extensive search for stationary states…”. In this point I am sympathetic with the authors, as there are no systematic solutions for this within this methodology. Molecular dynamics are probably the best answer, but it would not be fair to request that, as it represents another work. However, the authors certainly recognize that they cannot be sure that their extensive search didn’t miss a better way… So, at least a word of caution is recommended.

4 - This is just a point that muddled this review: the text mentions “product 2a” and table 1 includes 2a and 2b, while Scheme 1 labels products as 2 and 3. Please check consistency.

5 - Considering the reactions proflles (Fig 2 and 4) I find quit odd that the association of reactants (in the pre-reaction complex) result in an energy increase. In am quite sure that the electronic total energy of the complex is lower than the sum of the electronic energy of the reactants. So, this means that the negative entropy contribution to the Gibbs free energy is quite important. But entropy is not calculated in Gaussian without several assumptions (I refer the authors to the technical note “Thermochemistry in Gaussian” (Gaussian web site) to be aware of that). The authors should produce the same figures with electronic energy (with or without ZPVE correction) and include them, at least in the supplementary material.

6 – By comparing the Gibbs energies of activation for SN2 and E1 mechanisms (ca. 28 and 30 kcal/mol, respectively), the authors concluded their difference “accounts for the experimentally observed exclusive formation of SN2 product”. However, the difference of ca. 2 kcal/mol is within the error of most DFT functionals concerning reaction energies. Even if M06-2X provides a mean absolute error (please refer to benchmark studies) of 1 kcal/mol, this energy difference may be elusive. The authors must rephrase or fully justify this conclusion before acceptance of the manuscript.

Author Response

We thank the Reviewers for many helpful comments.

Reviewer 2

  1. The choice of M06-2X functional should be justified. This functional may be the best for this study of reaction mechanisms, but the authors must refer to at least one of the several benchmarking studies to justify their choice and set the limits of accuracy (see also comment 6 below).

We cite the following benchmark work:

E. G. Hohenstein, S. T. Chill,, C. D. Sherrill, Assessment of the Performance of the M05−2X and M06−2X Exchange-Correlation Functionals for Noncovalent Interactions in Biomolecules, J. Chem. Theory Comput. 2008, 4, 1996–2000.

  1. For the quantum chemical description of anions, the presence of diffuse functions is often imperative. Is not the case for F-? The authors should state why.

Including the diffuse functions usually increase the computational efforts very much. We think that the present methodology is optimal for accuracy vs. economy.

  1. The authors state “We carried out extensive search for stationary states…”. In this point I am sympathetic with the authors, as there are no systematic solutions for this within this methodology. Molecular dynamics are probably the best answer, but it would not be fair to request that, as it represents another work. However, the authors certainly recognize that they cannot be sure that their extensive search didn’t miss a better way… So, at least a word of caution is recommended.

Indeed, we found that there exists a structure for free reactant (TBMA+F-) whose Gibbs free energy is lower than that given in our previous manuscript. We carried out calculations based on this one, and the corresponding reaction mechanisms are updated in Figure 1-4. We revise the paragraph in Computational Details to:

We carried out extensive (but not exhaustive) search for stationary states over the potential energy surface of the system.

  1. This is just a point that muddled this review: the text mentions “product 2a” and table 1 includes 2a and 2b, while Scheme 1 labels products as 2 and 3. Please check consistency.

Corrections are made as requested.

  1. Considering the reactions proflles (Fig 2 and 4) I find quit odd that the association of reactants (in the pre-reaction complex) result in an energy increase. In am quite sure that the electronic total energy of the complex is lower than the sum of the electronic energy of the reactants. So, this means that the negative entropy contribution to the Gibbs free energy is quite important. But entropy is not calculated in Gaussian without several assumptions (I refer the authors to the technical note “Thermochemistry in Gaussian” (Gaussian web site) to be aware of that). The authors should produce the same figures with electronic energy (with or without ZPVE correction) and include them, at least in the supplementary material

In most SN2 reactions we have studied, the Gibbs free energies of pre-reaction complexes were lower than those of free reactants. There still exists some ambiguity concerning the existence of the pre-reaction complexes, and some theoretical studies suggested that pre-reaction complexes could not be obtained in some cases. We are not sure how the ‘well-skipping’ SN2 reactions predicted in the present work are related to those findings. Therefore, we add the following references, and a paragraph for discussion:

  1. Gao, Y. Zhao, Q. Guan, F. Wang, Ab initio kinetics predictions for the role of pre-reaction complexes in hydrogen abstraction from 2-butanone by OH radicals, RSC Adv. 2020, 10, 33205-33212.

  1. V. Dalessandro, J. R. Pliego Jr., Theoretical design of new macrocycles for nucleophilic fluorination with KF: thiourea-crown-ether is predicted to overcome [2.2.2]-cryptand, Mol. Syst. Des. Eng. 2020,

5, 1513-1523.

  1. Á. Dékány, G. Z. Kovács, G. Czakó, High-Level Systematic Ab Initio Comparison of Carbon- and Silicon-Centered SN2 Reactions, J. Phys. Chem. A 2021, 125, 9645–9657.

The relative Gibbs free energies of pre-reaction complexes in Figure 2 and Figure 4 with respective to those of free reactants are worth mentioning: The (stationary) pre-reaction complexes are predicted to lie higher than free reactants, in contrast with most SN2 reactions we have studied. Although some theoretical studies [41–43] suggested that pre-reaction complexes could not be obtained in some cases, the origin the ‘well-skipping’ SN2 reactions predicted here is not clear.

  1. By comparing the Gibbs energies of activation for SN2 and E1 mechanisms (ca. 28 and 30 kcal/mol, respectively), the authors concluded their difference “accounts for the experimentally observed exclusive formation of SN2 product”. However, the difference of ca. 2 kcal/mol is within the error of most DFT functionals concerning reaction energies. Even if M06-2X provides a mean absolute error (please refer to benchmark studies) of 1 kcal/mol, this energy difference may be elusive. The authors must rephrase or fully justify this conclusion before acceptance of the manuscript.

In our updated mechanisms, the E2 barriers are calculated to be higher (by 2.8 – 4.2 kcal/mol) than the SN2 barriers. This difference is well beyond the usual DFT error range (0.5 – 1 kcal/mol), thus corroborating our conclusion for the relative feasibility of SN2 vs. E2 reactions. We revise the paragraph to:

The Gibbs free energies of activation G for the elimination process are calculated to be 34.1 ~ 35.5 kcal/mol (Figure 4), which are considerably higher (by 2.8 – 4.2 kcal/mol) than those for the SN2 18F-fluorination (Figure 2), accounting for the experimentally observed exclusive formation of SN2 product.

Round 2

Reviewer 1 Report

The present form of the manuscript is adequate for publication.

Reviewer 2 Report

The authors' replies and corrections are sufficient to recommend acceptance of the manuscript in the present form.